

# Ground state topology of a four-terminal superconducting double quantum dot

Lev Teshler[1], Hannes Weisbrich[1], Jonathan Sturm[2],
Raffael L. Klees[2], Gianluca Rastelli[3] and Wolfgang Belzig[1]⋆

**1** Fachbereich Physik, Universität Konstanz, Universitätsstraße 40, 78457 Konstanz, Germany
**2** Institut für Theoretische Physik und Astrophysik,
Julius-Maximilians-Universität Würzburg,
Am Hubland, 97074 Würzburg, Germany
**3** Pitaevskii BEC Center, CNR-INO and Dipartimento di Fisica,
Università di Trento, 38123 Trento, Italy

⋆ wolfgang.belzig@uni-konstanz.de

## Abstract

In recent years, various classes of systems were proposed to realize topological states of matter. One of them are multiterminal Josephson junctions where topological Andreev bound states are constructed in the synthetic space of superconducting phases. Crucially, the topology in these systems results in a quantized transconductance between two of its terminals comparable to the quantum Hall effect. In this work, we study a double quantum dot with four superconducting terminals and show that it has an experimentally accessible topological regime in which the non-trivial topology can be measured. We also include Coulomb repulsion between electrons which is usually present in experiments and show how the topological region can be maximized in parameter space.



# 1   Introduction

Topology and related concepts is a central theme of modern solid state physics by explaining ultimately the nature of quantized phenomena. The discovery of the quantum Hall effect [1] and its quantized conductance which can be explained in terms of the 2D topological invariant of the first Chern number [2] started an ongoing search for novel topological materials for the past decades. It brought a unique understanding of topological insulators that are insulating in the bulk but have conducting edge states [3]. Moreover, the study of topological superconductors [4] started the concept of Majorana zero-energy modes that promises topologically protected quantum computing [5–8] on the basis of their non-Abelian braiding statistics.

   More recently, synthetic topological matter is intensely studied. Here the topology is constructed in internal degrees of freedom and thus allows for a better control of the topological properties. Prime examples are topological photonics [9, 10], topological driven Floquet systems [11] and topological electrical circuits [12]. This idea was also successfully transferred to multiterminal Josephson junctions that can host topological Andreev bound states in the space of superconducting phase differences [13–26] or topological superconducting circuits. [27–29]. With such approach the dimensionality of the phase space and the related topology is in principle unlimited and only depends on the number of superconducting terminals [30–32]. The non-trivial topology of the bound states results in a quantized transconductance [13] in units of $4e^2/h$ when applying voltages to the superconducting terminals. In general, these Andreev bound states can be also experimentally accessed by coherent microwave drive [33–35] and spectroscopy [36, 37]. Despite the fact that there are already experimental realizations of multiterminal Josephson junction devices [38–41], an evidence of a non-trivial topology in such system is still lacking. Although there are numerous proposals regarding topological multiterminal Josephson junctions these systems remain experimentally challenging due to the need of specific scattering regions in between the terminals [14–17]. An alternative approach is based on the coupling of quantum dots to superconducting leads [21, 22, 42] that in principle leads to a better control of the processes within the multiterminal Josephson junction. However the existing proposals still require an advanced coupling between the superconducting terminals that limits the feasibility of such proposals. Here we study a system with a double quantum dot that only requires a coupling between a double quantum dot and four superconducting terminals that hosts topological Andreev bound states, see Figure 1. Similar systems based on semiconductor nanowires with a double quantum dot coupled to two terminals [43, 44] or a single quantum dot coupled to three terminals [45] were already realized experimentally and, thus, this proposal should be in the reach of current experimental devices to finally observe topological Andreev bound states and the related quantized transconductance.

   We begin by deriving the effective low energy Hamiltonian of the system in the single-particle picture in section 2.1 and analyzing the topology of its Andreev bound states in section

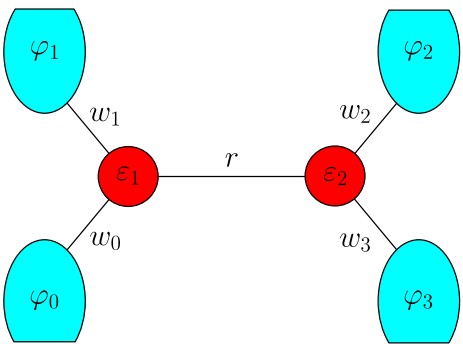

Figure 1: The system consists of a double quantum dot with inter-dot coupling $r$ and four superconductors with respective superconducting phases $\varphi_j$ which are coupled to the quantum dots via $w_j$ for $j = 0, 1, 2, 3$. The first two superconductors are coupled to the first quantum dot and the last two superconductors are coupled to the second quantum dot.

2.2. In Section 2.3, we transition to the many-body picture, enabling the incorporation of the Coulomb interaction — a critical factor for comprehending real devices. While the Coulomb interaction has been extensively investigated in quantum dot junctions in general, e.g. in [46–52], this section exclusively focuses on its implications for the topology of bound states. Finally, in section 2.4 we look at the total spin of the many-body ground state and in section 2.5 discuss the parameter regimes for which the non-trivial topology can be detected.

## 2 Main results

### 2.1 Single-particle Hamiltonian

The considered system is a double quantum dot (DQD), where each quantum dot is coupled to two superconductors (SCs) via couplings $w_j$ (see Figure 1). In the absence of any coupling between the quantum dots it describes two isolated Josephson junctions (JJs) whose Andreev bound states (ABS) have energies $\pm E_1(\varphi_1 - \varphi_0)$ and $\pm E_2(\varphi_3 - \varphi_2)$, with $\varphi_i$ ($i = 0, 1, 2, 3$) being the phases of the superconducting terminals. However, for finite coupling $r$ between the quantum dots the ABS hybridize and, thus, depend on all SC phases. This hybridization is crucial for the non-trivial topology of the bound states in the SC phase space as will be discussed below.

The Hamiltonian of the DQD system can be be written in the basis $\tilde{d}^\dagger = (d_{1,\uparrow}^\dagger, d_{1,\downarrow}, d_{2,\uparrow}^\dagger, d_{2,\downarrow})$ of fermionic operators in the dot-spin space as $H_D = \tilde{d}^\dagger \tilde{H}_D \tilde{d}$ with the matrix Hamiltonian

$$\tilde{H}_D = \begin{pmatrix} \varepsilon_1 & r \\ r & \varepsilon_2 \end{pmatrix} \otimes \tau_3 \, . \tag{1}$$

Here, $\varepsilon_1$, $\varepsilon_2$ are the on-site energies of the dots, $r$ the inter-dot coupling and $\tau_j$ the Pauli matrices.

The four superconductors, in turn, are described by $H_S = \sum_k \tilde{c}_k^\dagger \tilde{H}_{S,k} \tilde{c}_k$ with $\tilde{c}_k^\dagger = (c_{0k\uparrow}^\dagger, c_{0-k\downarrow}, \ldots, c_{3k\uparrow}^\dagger, c_{3-k\downarrow})$ with matrix Hamiltonian $\tilde{H}_{S,k} = \mathrm{diag}(\tilde{H}_{S,0k}, \tilde{H}_{S,1k}, \tilde{H}_{S,2k}, \tilde{H}_{S,3k})$, where each block on the diagonal,

$$\tilde{H}_{S,jk} = \xi_{jk} \tau_3 + \Delta_j e^{i\varphi_j \tau_3} \tau_1 \, , \tag{2}$$

describes a BCS superconductor with superconducting phase $\varphi_j$, gap $\Delta_j$ and normal state dispersion $\xi_{jk}$. In the following, we choose a gauge such that $\varphi_0 = 0$.

To describe the coupling between the DQD and the SC terminals, we introduce $H_{DS} = \sum_k (\tilde{d}^\dagger \tilde{V}_{DS} \tilde{c}_k + \text{H.c.})$ with

$$\tilde{V}_{DS} = \begin{pmatrix} w_0 & w_1 & 0 & 0 \\ 0 & 0 & w_2 & w_3 \end{pmatrix} \otimes \tau_3 \,, \tag{3}$$

where $w_j$ is the coupling of terminal $j$ to the quantum dots.

The Dyson equation for the total system (see Appendix A) can be solved for the dressed Green's function (GF) of the DQD, $\tilde{G}_{DD} = \tilde{g}_{DD} + \tilde{g}_{DD} \tilde{\Sigma}_{DD} \tilde{G}_{DD}$, where we identified the self-energy $\tilde{\Sigma}_{DD} = \tilde{V}_{DS} \tilde{g}_{SS} \tilde{V}_{DS}^\dagger$ that describes the proximity coupling of the DQD to the SCs. Here, $\tilde{G}_i$ and $\tilde{g}_i$ are the dressed and bare matrix GF, respectively, and $i = D, S$ stands for the DQD and SC system, respectively. To calculate the self-energy, we need the bare GF of the SC system, which we obtain by inverting its Hamiltonian as $\tilde{g}_{SS}(E) = (E - \tilde{H}_S)^{-1}$.

Assuming the energy $E$ to be much smaller than the superconducting gaps $\Delta_j$ of the superconductors (low energy limit), we obtain the matrix GF $\tilde{g}_{SS} = \text{diag}(\tilde{g}_{S,0}, \tilde{g}_{S,1}, \tilde{g}_{S,2}, \tilde{g}_{S,3})$ with

$$\tilde{g}_{S,j}(E) = -\pi N_0 \frac{E + \Delta_j e^{i\varphi_j \tau_3} \tau_1}{\sqrt{\Delta_j^2 - E^2}} \xrightarrow{E \to 0} -\pi N_0 e^{i\varphi_j \tau_3} \tau_1 \,, \tag{4}$$

for $j = 0, 1, 2, 3$. $N_0$ is the density of states at the Fermi energy in the leads. A detailed calculation of this can be found in [22].

Calculating the self-energy (Eq. (A.3)) and adding it to the Hamiltonian in Eq. (1) we obtain the effective low energy Hamiltonian of the DQD in the spinor basis $\tilde{d}^\dagger = \left( d_{1,\uparrow}^\dagger, d_{1,\downarrow}, d_{2,\uparrow}^\dagger, d_{2,\downarrow} \right)$,

$$\tilde{H}_{D,\text{eff}} = \begin{pmatrix} \varepsilon_1 & \Gamma_0 + \Gamma_1 e^{i\varphi_1} & r & 0 \\ \Gamma_0 + \Gamma_1 e^{-i\varphi_1} & -\varepsilon_1 & 0 & -r \\ r & 0 & \varepsilon_2 & \Gamma_2 e^{i\varphi_2} + \Gamma_3 e^{i\varphi_3} \\ 0 & -r & \Gamma_2 e^{-i\varphi_2} + \Gamma_3 e^{-i\varphi_3} & -\varepsilon_2 \end{pmatrix}, \tag{5}$$

where $\Gamma_j = \pi N_0 w_j^2$ are the effective couplings between the quantum dots and the superconducting terminals. The self-energy terms can be interpreted as coherent tunneling of Cooper pairs between superconductors and dots with phase-dependent amplitudes that we are going to denote $\Gamma_{01} = \Gamma_0 + \Gamma_1 e^{i\varphi_1}$ and $\Gamma_{23} = \Gamma_2 e^{i\varphi_2} + \Gamma_3 e^{i\varphi_3}$ for later convenience.

## 2.2 Andreev bound states and topology

The DQD system has four ABS (see Figure 2), as follows from the $4 \times 4$ low energy Hamiltonian in Eq. (5). Their energies can be calculated by imposing $\det(E - H_{D,\text{eff}}) = 0$ which leads to a quartic equation with solutions,

$$E_{1,2}^2 = b/2 \pm \sqrt{(b/2)^2 - c} \,, \tag{6}$$

where

$$\begin{aligned} b &= \varepsilon_1^2 + \varepsilon_2^2 + 2r^2 + |\Gamma_{01}|^2 + |\Gamma_{23}|^2 \,, \\ c &= r^4 + \varepsilon_1^2 \varepsilon_2^2 + \varepsilon_1^2 |\Gamma_{23}|^2 + \varepsilon_2^2 |\Gamma_{01}|^2 + |\Gamma_{01}|^2 |\Gamma_{23}|^2 + r^2 (\Gamma_{01} \Gamma_{23}^* + \Gamma_{23} \Gamma_{01}^* - 2\varepsilon_1 \varepsilon_2) \,. \end{aligned} \tag{7}$$

The four eigenenergies come in symmetric pairs $\pm E_{1,2}$ as a result of particle-hole (PH) symmetry of the system. For $r = 0$, the system decouples into two independent Josephson junctions with eigenenergies $E_1 = \sqrt{\varepsilon_1^2 + |\Gamma_{01}|^2}$ and $E_2 = \sqrt{\varepsilon_2^2 + |\Gamma_{23}|^2}$. The isolated ABS each depend only on one phase difference and are, hence, topologically trivial.

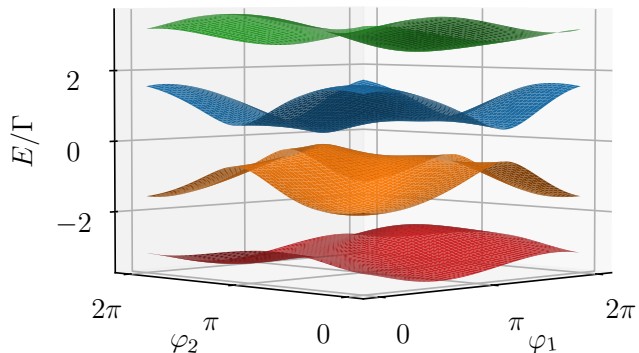

Figure 2: All four energy bands of the two-dot system for $\varphi_3 = \pi/2$, $\varepsilon_i = \Gamma_i = \Gamma$ and $r = 1.5\Gamma$. PH symmetry results in a symmetric spectrum. In the ground state only the red and orange energy bands with negative energy are occupied.

Generically, the Hamiltonian in Eq. (5) and its eigenstates depend on three phase differences which serve as synthetic dimensions for the topology. A non-trivial topology in terms of the GS Chern number $C_{12}^{\text{GS}}$ with respect to the superconducting phases $\varphi_1$ and $\varphi_2$ leads to a quantized transconductance $G_{12} = (-4e^2/h)C_{12}^{\text{GS}}$ between terminals 1 and 2 when applying incommensurate voltages to those two terminals [13]. Here, $e$ is the elementary charge and $h$ the Planck constant.

With $|n\rangle$ denoting the phase-dependent eigenstates sorted by $n = 1, 2, 3, 4$ in order of increasing energy, the Berry Connection of the $n$th eigenstate is defined as $a_\alpha^{(n)} = i \langle n| \partial_\alpha |n\rangle$ where we use $\partial_\alpha = \partial/\partial\varphi_\alpha$. From this we can define the Berry curvature $f_{\alpha\beta}^{(n)} = \partial_\alpha a_\beta^{(n)} - \partial_\beta a_\alpha^{(n)}$ of the $n$th eigenstate with respect to the phases $\varphi_\alpha$ and $\varphi_\beta$. Finally, the corresponding Chern number of the $n$th eigenstate is defined as

$$c_{\alpha\beta}^{(n)} = \frac{1}{2\pi} \int_0^{2\pi} \int_0^{2\pi} f_{\alpha\beta}^{(n)} \, \mathrm{d}\varphi_\alpha \mathrm{d}\varphi_\beta \,, \tag{8}$$

which is quantized to integer values. Since we are in the single-particle picture, in the ground state (GS) all states below the Fermi energy ($E_F = 0$) are occupied. Because of PH symmetry of the Hamiltonian, two of the four eigenenergies are negative, meaning that the ground state Chern number is given by the sum over the two lowest eigenstates,

$$C_{12}^{\text{GS}} = c_{12}^{(1)} + c_{12}^{(2)} \,. \tag{9}$$

Changing the GS Chern number requires a closing of the gap between two energy bands at zero energy, so called Weyl points, which change the value of the Chern number by their topological charge equal to $\pm 1$. Figures 3b and 3c show such Weyl points in $\varphi_1$-$\varphi_2$ space. The two spectra are obtained for parameters corresponding to the red and blue dots in Figure 3a, respectively, which lie on boundaries between different topological phases. In the symmetric case ($\Gamma_i = \Gamma$ and $\varepsilon_i = \varepsilon$) the topological phases are bounded by $r = |\varepsilon|$, $\varphi_3 = \pi$ and $r(\varphi_3) = \sqrt{\varepsilon^2 + 4\Gamma^2 \sin^2(\varphi_3/2)}$ along which Weyl points occur, as we show in Appendix B. The total area of the non-trivial topological phase is, thus, given by the integral

$$A_{\text{top}} = \int_0^{2\pi} \left[ \sqrt{\varepsilon^2 + 4\Gamma^2 \sin^2(\varphi_3/2)} - |\varepsilon| \right] \mathrm{d}\varphi_3 \,, \tag{10}$$

which is maximized for $\varepsilon = 0$ (see Figure 6). Remarkably, a finite topological region appears for any $|\Gamma| \neq 0$, while for $\Gamma = 0$ no topological effect is possible since the SCs are completely

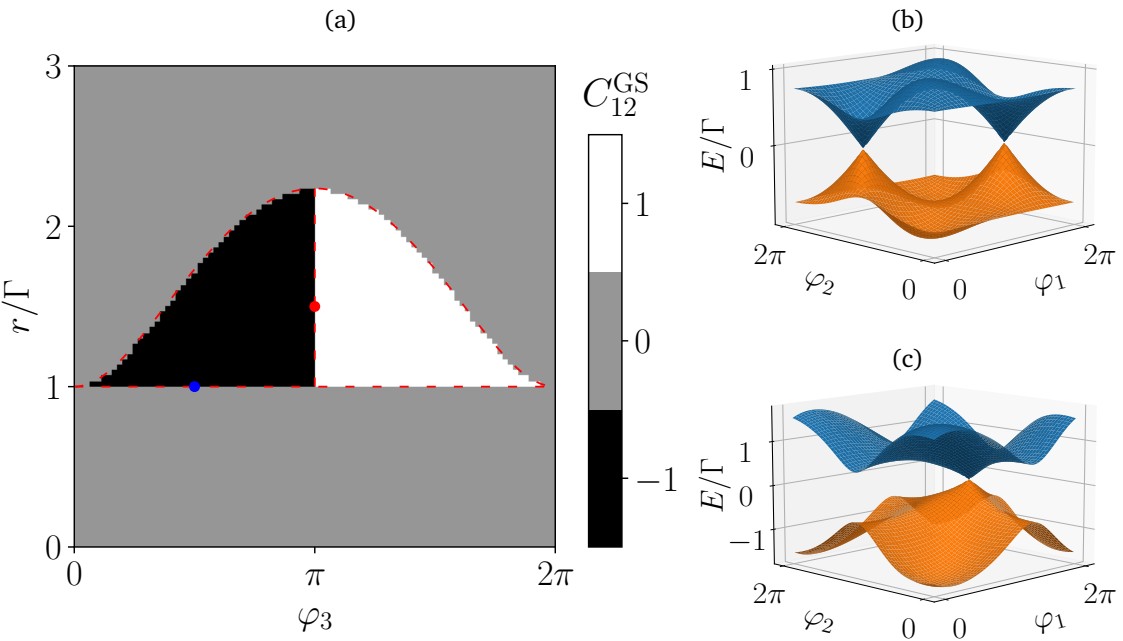

Figure 3: (a) Topological phase diagram: First Chern number as a function of $r$ and $\varphi_3$ for $\Gamma_i = \Gamma = \varepsilon_i$. The dashed red lines indicate the analytically calculated boundaries of the topological phases. (b) Spectrum of the ground state and first excited state for $\varphi_3 = \pi$ and $r = 1.5\Gamma$ (red point). (c) Same spectrum for $\varphi_3 = \pi/2$ and $r = \Gamma$ (blue point).

decoupled. In the case of asymmetric parameters it is more complicated to find an analytical formula. We can, however, calculate the topological phase diagram numerically for some values to estimate the effect qualitatively. This is shown in Figure 7. We find that asymmetric parameters do not increase of the topological region and in most cases even reduce it.

## 2.3 Many-body Hamiltonian

The Hamiltonian in Eq. (5) is written in the basis $\tilde{d}^\dagger$ of single-particle (SP) creation and annihilation operators. Thus, treatment in the single-particle picture is sufficient. To describe the Zeeman splitting in a magnetic field $B$ and the Coulomb repulsion between two electrons, however, we extend our Hamiltonian to

$$H_{\mathrm{D}} = H_{D,\mathrm{eff}} + \frac{B}{2}(n_{1\uparrow} + n_{2\uparrow} - n_{1\downarrow} - n_{2\downarrow}) + U_C(n_{1\uparrow}n_{1\downarrow} + n_{2\uparrow}n_{2\downarrow}), \tag{11}$$

where $n_{i\sigma} = d_{i\sigma}^\dagger d_{i\sigma}$ are the particle number operators for electrons with spin $\sigma$ on dot $i$ and $U_C$ is the Coulomb energy of an electron pair on the same dot. We only include Coulomb interaction between electrons on the same dot (intra-dot), since the inter-dot Coulomb interaction should be significantly weaker and, if included, has a very similar effect on the topological phase diagram. In the following, we assume $U_C, B \ll \Delta$, which simplifies the consideration significantly.

The Hamiltonian in Eq. (11) can not be expressed in the single-particle basis $\tilde{d}^\dagger$. We, therefore, need to choose an appropriate basis of many-body (MB) states. In the many-body picture, in the GS only the lowest energy state is occupied, so the GS Chern number is just the Chern number of the lowest energy eigenstate. For $B = U_C = 0$ this Chern number coincides

with the GS Chern number in the single-particle picture from Eq. (9), as the two pictures become equivalent. The Zeeman term, however, could in principle also be included in a SP Hamiltonian. For $U_C \neq 0$, the Chern number of the MB ground state might change and has to be calculated separately.

First, let us note that each of the two quantum dots can host zero, one or two electrons, in the latter case with opposite spin due to the Pauli principle. The possible single quantum dot states are, thus, $|0\rangle, |\uparrow\rangle, |\downarrow\rangle, |\uparrow\downarrow\rangle$. For the double quantum dot, we have $2^4 = 16$ possible basis states, which we denote as $|\alpha, \beta\rangle$, if dot 1 is in state $|\alpha\rangle$ and dot 2 in state $|\beta\rangle$. We would like to note, that the Hamiltonian in Eq.(11) contains only terms with an even number (two or four) of fermionic operators and, thus, conserves the fermionic parity. We can, thus, divide the many-body states into two sectors with even and odd numbers of electrons. Conservation of the total spin allows us to subdivide the even parity sector into a singlet (spin 0) and triplet (spin 1) sector, while the odd parity sector corresponds to a doublet with spin 1/2. We shall look at the 3 spin sectors separately.

For the singlet sector, we can choose the basis $\{|0,0\rangle, (|\uparrow,\downarrow\rangle - |\downarrow,\uparrow\rangle)/\sqrt{2}, |\uparrow\downarrow,0\rangle, |0,\uparrow\downarrow\rangle, |\uparrow\downarrow,\uparrow\downarrow\rangle\}$ and obtain the matrix Hamiltonian

$$\tilde{H}_D^{\text{singlet}} = \begin{pmatrix} 0 & 0 & \Gamma_{01}^* & \Gamma_{23}^* & 0 \\ 0 & \varepsilon_1 + \varepsilon_2 & \sqrt{2}r & \sqrt{2}r & 0 \\ \Gamma_{01} & \sqrt{2}r & 2\varepsilon_1 + U_C & 0 & \Gamma_{23}^* \\ \Gamma_{23} & \sqrt{2}r & 0 & 2\varepsilon_2 + U_C & \Gamma_{01}^* \\ 0 & 0 & \Gamma_{23} & \Gamma_{01} & 2(\varepsilon_1 + \varepsilon_2) + 2U_C \end{pmatrix}. \tag{12}$$

How the many-body Hamiltonian is calculated is explained in Appendix C. This $5 \times 5$ Hamiltonian has five energy levels, one of which is completely phase-independent due to symmetry. The doublet sector has eight energy levels. As basis we set $|\sigma\pm\rangle = (|\sigma,0\rangle \pm |0,\sigma\rangle)/\sqrt{2}$ and $|t\sigma\pm\rangle = (|\sigma,\uparrow\downarrow\rangle \pm |\uparrow\downarrow,\sigma\rangle)/\sqrt{2}$ for $\sigma = \uparrow, \downarrow$. Choosing the basis order $|\uparrow\pm\rangle, |t\uparrow\pm\rangle, |\downarrow\pm\rangle, |t\downarrow\pm\rangle$, we obtain the matrix Hamiltonian $\tilde{H}_D^{\text{doublet}} = \sigma_0 \otimes \tilde{H}_{\text{block}} + (B/2)\sigma_3 \otimes \mathbb{1}_{4\times4}$, where the last term describes the spin-dependent Zeeman splitting and

$$\tilde{H}_{\text{block}} = \frac{1}{2}\begin{pmatrix} \varepsilon_1 + \varepsilon_2 + 2r & \varepsilon_1 - \varepsilon_2 & \Gamma_{01}^* + \Gamma_{23}^* & \Gamma_{01}^* + \Gamma_{23}^* \\ \varepsilon_1 - \varepsilon_2 & \varepsilon_1 + \varepsilon_2 - 2r & \Gamma_{01}^* + \Gamma_{23}^* & \Gamma_{01}^* + \Gamma_{23}^* \\ \Gamma_{01} + \Gamma_{23} & \Gamma_{01} + \Gamma_{23} & 3\varepsilon_1 + 3\varepsilon_2 - 2r + 2U_C & \varepsilon_2 - \varepsilon_1 \\ \Gamma_{01} + \Gamma_{23} & \Gamma_{01} + \Gamma_{23} & \varepsilon_2 - \varepsilon_1 & 3\varepsilon_1 + 3\varepsilon_2 + 2r + 2U_C \end{pmatrix}, \tag{13}$$

is the same for both spin subspaces.

Finally, we can choose $|\uparrow,\uparrow\rangle, (|\uparrow,\downarrow\rangle + |\downarrow,\uparrow\rangle)/\sqrt{2}$ and $|\downarrow,\downarrow\rangle$ as basis of the triplet space. The corresponding Hamiltonian is found to be diagonal with phase-independent eigenenergies $\varepsilon_1 + \varepsilon_2(\pm B)$. These triplet states are, obviously, topologically trivial, since we are considering spin-singlet superconductors which do not couple to triplet states in the double dot. For the doublet sector the analysis is more intricate. Its levels depend on the SC phase differences and could, in principle, be topological. Because of the high computational effort we could not calculate the Chern numbers in the entire (even reasonably restricted) parameter space. Instead, we looked for Weyl points by plotting the minimal energy gap between the GS and first excited state of the doublet sector minimizing it over a large parameter region. The only point where the energy gap closes is for $\varepsilon = U_C = 0$ (see Figure 8 in Appendix E). We can, therefore, restrict the search for a topological phase transition to this point. We find that for $\varepsilon = U_C = 0$ two Weyl points appear for all values of $\varphi_3$ and $r$ making the Chern number ill-defined. Outside of this point, however, the Chern number stays zero (both for positive and negative $\varepsilon$ and $U_C$). This can be explained by the fact that the two Weyl points occuring in the transition have opposite topological charges which cancels their effect on the Chern number. Based on this analysis, we conclude that the doublet GS has no finite topological phase. The

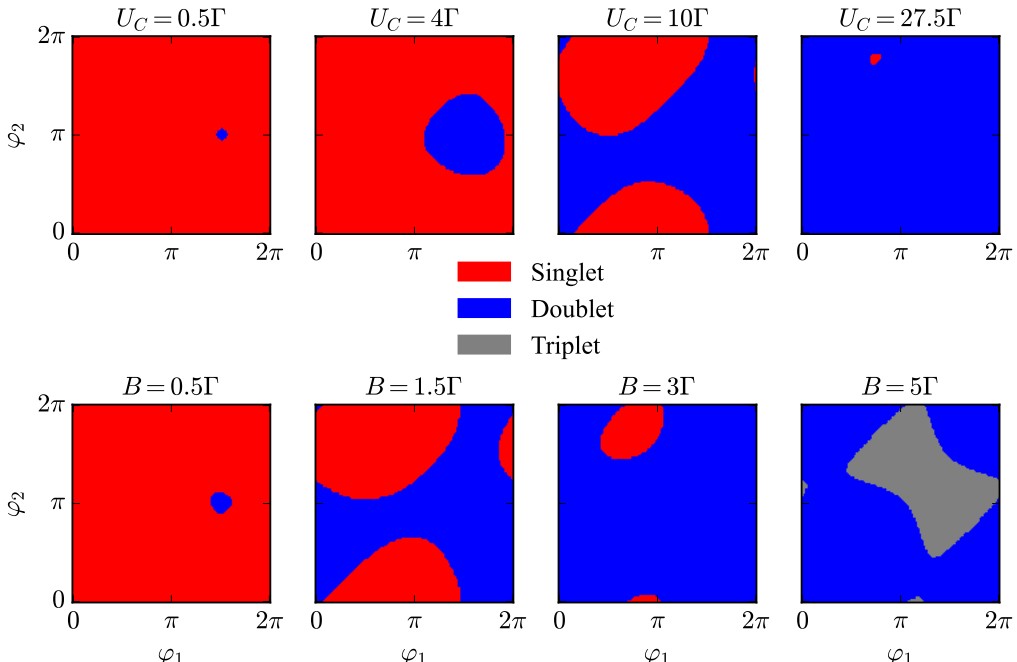

Figure 4: Ground state spin phase diagrams in the $\varphi_1$-$\varphi_2$ space for $\varepsilon_i = \Gamma_i = \Gamma$, $r = 1.2\Gamma$ and $\varphi_3 = \pi/2$. In the upper row, the Coulomb interaction $U_C$ is varied, while $B = 0$, in the lower row $U_C = 0$ and the magnetic field $B$ is varied.

regime $\varepsilon = U_C = 0$ for which the Chern number is ill-defined is not experimentally relevant as it is not robust against fluctuations in those parameters and, thus, not observable. This being said, we can, in the following, focus on the singlet sector, as only its states can be topologically non-trivial. It is, however, only relevant, as long the singlet GS is the overall GS which is crucial for the observation of a quantized transconductance.

## 2.4 Ground state spin

To find the relevant conditions under which the system has a potentially topological GS, we identify the spin of the overall GS simply by comparing the ground states of the three spin subspaces. Having already found non-trivial topology in the absence of Coulomb interaction and magnetic fields in section 2.2, we will now look at the effect of these two parameters on the GS spin. Fixing $\varepsilon_i = \Gamma_i = \Gamma$, $r = 1.2\Gamma$ and $\varphi_3 = \pi/2$ (values, for which we previously found non-trivial topology), we plot the spin of the GS in the $\varphi_1$-$\varphi_2$ Brillouin zone for different values of $U_C$ and $B$. This is shown in Figure 4. For weak Coulomb repulsion and magnetic fields, we have a singlet GS (red). For $U_C > 0.5\Gamma$, the doublet GS (blue) gradually takes over the entire Brillouin zone. Similarly, with increasing $B$, the Zeeman splitting of the doublet GS eventually overcomes the singlet GS which is unaffected by $B$. For strong magnetic fields, the triplet GS (gray) prevails, since it has the strongest Zeeman splitting. We conclude that in order to have a topological GS and exactly quantized transconductance, we have to avoid magnetic fields and minimize the Coulomb repulsion between electrons on the same dot.

## 2.5 Enhancement of the topological phases

For practical applications it is convenient to make the topological phases as large as possible in parameter space, making the quantized conductance $G_{12}$ more robust against fluctuations in those parameters. At the same time, observing the quantization also requires a finite energy

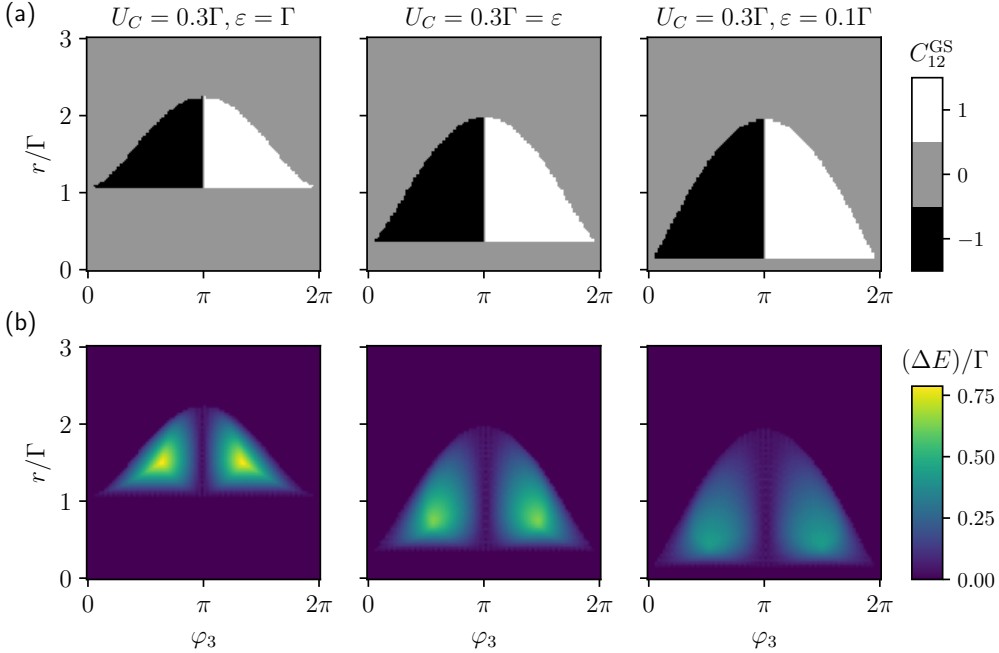

Figure 5: (a) Ground state Chern number $C_{12}$ as a function of $\varphi_3$ and $r$ for $U_C = 0.3\Gamma$ and decreasing dot energies $\varepsilon = \Gamma, 0.3\Gamma, 0.1\Gamma$ from left to right. (b) The energy gap $(\Delta E)/\Gamma$ between ground state and first excited state in the region of non-zero Chern number from the corresponding phase diagrams above in (a).

gap separating the GS from the other energy levels. While it is realistic to assume $B \ll \Gamma_i$, it might be problematic to achieve $U_C \ll \Gamma_i$ in experiments. We, thus, assume a finite value $U_C = 0.3\Gamma_0$, which lies in the allowed range $0 < U_C < 0.5\Gamma$ and try to find the values for $\varepsilon_i$ and $\Gamma_i$ that maximize the area of the non-trivial topological phases in $\varphi_3$-$r$ space. Comparing the phase diagram for $U_C = 0.3\Gamma$ and $U_C = 0$, we see only a small shift of the lower boundary towards higher values of $r$. This can be understood if one notes that the Coulomb energy essentially shifts the dot energies from $\varepsilon$ to $\varepsilon + U_C$, such that the lower boundary of the topological region is also shifted up, as we showed in the SP picture. This explains the effect of $U_C$ on the phase diagram, at least qualitatively. Besides this small shift, the phase diagram of the MB singlet GS retains all the dependencies found for the SP GS: the topological region is the largest for symmetric dot energies $\varepsilon_i = \varepsilon$ and SC couplings $\Gamma_i = \Gamma$ and increases as $\varepsilon$ approaches zero, which can be seen in Figure 5(a). At the same time, the energy gap $\Delta E$ decreases, but still remains finite ($> 0.3\Gamma$) in a larger region even for very small $\varepsilon$ (Figure 5(b)). Interestingly, the gap increases with increasing $0 < U_C < 0.3\Gamma$ (Figure 9). On one hand, a larger gap means that the system stays in the GS for faster parameter evolution (according to the adiabatic theorem). On the other hand, $U_C$ has to be small to ensure the singlet GS is the overall GS of the system. Ultimately, a trade off between these two effects will determine the optimal $U_C$. This will likely depend on the details of the experimental setup which goes beyond the scope of this work.

## 3 Conclusion and outlook

In this paper, we found that the four-terminal superconducting DQD has a topological singlet ground state with non-trivial topology in terms of the first Chern number $C_{12}$. This topological

property could be in principle probed by measuring the transconductance when applying two voltages to terminals 1 and 2. As discussed in the manuscript, the topological phases exist in large parameter regions and are, thus, robust against fluctuations and noise.

We derived the effective single-particle Hamiltonian in the low energy limit and calculated its energy bands and toplogical phase diagram analytically. Introducing a many-body Hamiltonian allowed us to describe Coulomb repulsion between electrons on the same dot and determine the total spin of the ground state. We found that the topological phase can be maximized by choosing $\Gamma_i = \Gamma$ and $\varepsilon = \varepsilon_i = E_F = 0$. The energy gap, however, decreases with decreasing $\varepsilon$ which can become relevant in experiments. The non-trivial topology is preserved for finite Coulomb energy and magnetic field, but disappears when these quantities reach a critical value of about $\Gamma/2$.

We emphasize that this work discusses the relevant parameter regimes for which a quantized transconductance can be observed under experimentally realistic constraints of Coulomb interaction of a double quantum dot. Moreover, the proposed topological multi-terminal junction lies within the reach of current or near term double quantum dot devices.

As an outlook, we would like to mention that a possible spin-orbit coupling in such devices (which we neglected here) can have significant effect on the topology due to a further spin splitting of the bound states and thus directly influencing the discussed topology of the system. Besides, spin-orbit coupling could drive the system into the regime of a topological superconductor with possible Majorana states which would be distinct to the discussed synthetic topology of the bound states in the space of the superconducting phases.

## Acknowledgments

We acknowledge useful discussions with J. C. Cuevas.

**Funding information** This work was financially supported by the Deutsche Forschungsgemeinschaft (DFG; German Research Foundation) project no. 467596333 and project no. 425217212 via the Collaborative Research Center SFB 1432.

## A   Full Dyson equation and self-energy of the DQD

The full Dyson equation for the total system, consisting of the DQD and the four superconductors with coupling $V$ can be written in matrix form as

$$\begin{pmatrix} \tilde{G}_{DD} & \tilde{G}_{DS} \\ \tilde{G}_{SD} & \tilde{G}_{SS} \end{pmatrix} = \begin{pmatrix} \tilde{g}_D & 0 \\ 0 & \tilde{g}_S \end{pmatrix} + \begin{pmatrix} \tilde{g}_D & 0 \\ 0 & \tilde{g}_S \end{pmatrix} \begin{pmatrix} 0 & \tilde{V}_{DS} \\ \tilde{V}_{DS}^\dagger & 0 \end{pmatrix} \begin{pmatrix} \tilde{G}_{DD} & \tilde{G}_{DS} \\ \tilde{G}_{SD} & \tilde{G}_{SS} \end{pmatrix}. \tag{A.1}$$

Plugging the equation for the $G_{SD}$ component into the equation for $G_{DD}$, we obtain

$$\tilde{G}_{DD} = \tilde{g}_{DD} + \tilde{g}_{DD} \tilde{V}_{DS} \tilde{g}_{SS} \tilde{V}_{DS}^\dagger \tilde{G}_{DD}. \tag{A.2}$$

Comparing this with $\tilde{G}_{DD} = \tilde{g}_{DD} + \tilde{g}_{DD} \tilde{\Sigma}_{DD} \tilde{G}_{DD}$, yields the self-energy of the dot due to the dot-superconductor couplings, $\tilde{\Sigma}_{DD} = \tilde{V}_{DS} \tilde{g}_{SS} \tilde{V}_{DS}^\dagger$. Using the bare GF of the superconductors,

$\tilde{g}_{S,j}(E) = -\pi N_0 e^{i\varphi_j \tau_3} \tau_1$ and the identity $\tau_3 \tau_1 \tau_3 = -\tau_1$, we get

$$\tilde{\Sigma}_{DD} = \begin{pmatrix} w_0 & w_1 & 0 & 0 \\ 0 & 0 & w_2 & w_3 \end{pmatrix} \otimes \tau_3 \begin{pmatrix} \tilde{g}_{S,0} & 0 & 0 & 0 \\ 0 & \tilde{g}_{S,1} & 0 & 0 \\ 0 & 0 & \tilde{g}_{S,2} & 0 \\ 0 & 0 & 0 & \tilde{g}_{S,3} \end{pmatrix} \begin{pmatrix} w_0 & 0 \\ w_1 & 0 \\ 0 & w_2 \\ 0 & w_3 \end{pmatrix} \otimes \tau_3$$

$$= \begin{pmatrix} w_0^2 \tau_3 \tilde{g}_{S,0} \tau_3 + w_1^2 \tau_3 \tilde{g}_{S,1} \tau_3 & 0 \\ 0 & w_2^2 \tau_3 \tilde{g}_{S,2} \tau_3 + w_3^2 \tau_3 \tilde{g}_{S,3} \tau_3 \end{pmatrix} \tag{A.3}$$

$$= \begin{pmatrix} 0 & \Gamma_0 + \Gamma_1 e^{i\varphi_1} & 0 & 0 \\ \Gamma_0 + \Gamma_1 e^{-i\varphi_1} & 0 & 0 & 0 \\ 0 & 0 & 0 & \Gamma_2 e^{i\varphi_2} + \Gamma_3 e^{i\varphi_3} \\ 0 & 0 & \Gamma_2 e^{-i\varphi_2} + \Gamma_3 e^{-i\varphi_3} & 0 \end{pmatrix},$$

with $\Gamma_j = \pi N_0 w_j^2$.

# B  Condition for Weyl points

The condition for Weyl points is $c = 0$ with $c$ from Eq. (7), which for $\varepsilon_1 \varepsilon_2 \geq 0$ reduces to

$$(r^2 - |\varepsilon_1 \varepsilon_2| - |\Gamma_{01} \Gamma_{23}|)^2 + (|\varepsilon_1 \Gamma_{23}| - |\varepsilon_2 \Gamma_{01}|)^2 + 4r^2 |\Gamma_{01} \Gamma_{23}| \cos^2(\psi/2), \tag{B.1}$$

with $\psi = \arg(\Gamma_{01} \Gamma_{23}^*)$, which is the sum of three non-negative numbers, which all have to be zero to fulfill $c = 0$. The resulting three conditions are

$$r^2 = |\varepsilon_1 \varepsilon_2| + |\Gamma_{01} \Gamma_{23}|, \tag{B.2}$$

$$|\varepsilon_1 \Gamma_{23}| = |\varepsilon_2 \Gamma_{01}|, \tag{B.3}$$

$$\cos(\psi/2) = 0. \tag{B.4}$$

For $\Gamma_i = \Gamma$ and $\varepsilon_i = \varepsilon$ condition (B.3) simplifies to $\pm \varphi_1 = \varphi_2 - \varphi_3 + 2\pi n$, $n \in \mathbb{Z}$(*). On the other hand, plugging (B.3) into (B.2) leads to

$$r^2 = \varepsilon^2 + 4\Gamma^2 \cos^2 \left( \frac{\varphi_2 - \varphi_3}{2} \right). \tag{B.5}$$

This means that Weyl points are restricted to the the region $|\varepsilon| < r < \sqrt{\varepsilon^2 + 4\Gamma^2}$. With the notation $\Gamma_{01} \Gamma_{23}^* = a + ib$ ($a, b \in \mathbb{R}$) condition (B.4) is equivalent to $b = 0$ and $a \leq 0$. Writing out $a$ and $b$ and eliminating $\varphi_1$ by means of (*) leads to

$$2\sin(\varphi_3) + \sin(\varphi_2) + \sin(2\varphi_3 - \varphi_2) = 0, \tag{B.6}$$

$$2\cos(\varphi_3) + \cos(\varphi_2) + \cos(2\varphi_3 - \varphi_2) \leq 0. \tag{B.7}$$

One solution for these is $\varphi_2 = \varphi_3 \pm \pi$, while for $\varphi_3 = \pi$ any $\varphi_2$ is a solution. Combining this with (B.5) shows that Weyl points occur along the straight lines $r = |\varepsilon|$ and $\varphi_3 = \pi$ intersected with region $|\varepsilon| < r < \sqrt{\varepsilon^2 + 4\Gamma^2}$. Since the system is symmetric under exchange of $\varphi_2$ and $\varphi_3$, $\varphi_2 = \pi$ is also a solution for any $\varphi_3$, which together with (B.5) gives the curve

$$r(\varphi_3) = \sqrt{\varepsilon^2 + 4\Gamma^2 \sin^2(\varphi_3/2)}. \tag{B.8}$$

These are the only solutions for Weyl points. Since Weyl points occur when the Chern number changes, this explains the exact shape of the topological phase diagram of the single-particle GS in the symmetric case $\Gamma_i = \Gamma$, $\varepsilon_i = \varepsilon$. One particular observation is that a topological region appears for any $|\Gamma \neq 0|$. For $\Gamma = 0$, however, Eq. (B.8) becomes $r(\varphi_3) = |\varepsilon|$ which coincides with the other boundary and, therefore, encloses no region.

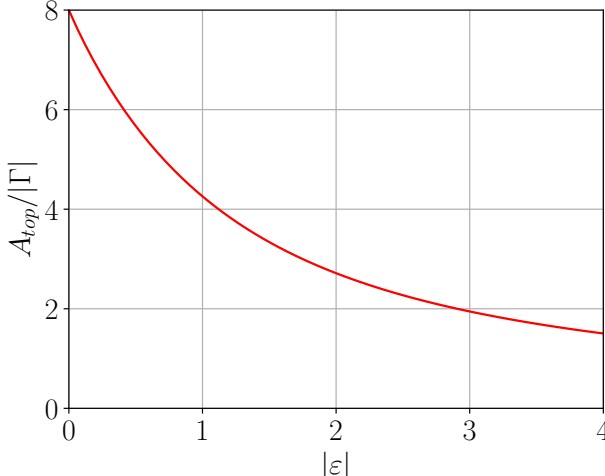

Figure 6: Area $A_{\text{top}}$ of the topological region in $\varphi_3$-$r$ space (in units of rad $\cdot |\Gamma|$) as a function of the dot energy $|\varepsilon|$ for the symmetric case $\varepsilon_i = \varepsilon$ and $\Gamma_j = \Gamma$. It is clearly a monotonically decreasing function and the maximum area is found for $\varepsilon = 0$.

## C   Calculation of the many-body Hamiltonian

Choosing a basis $B$ of many-body states, we obtain the matrix Hamiltonian $\tilde{H}$ of a Hamiltonian $H$ in second quantization by using twice the completeness relation $\sum_{\psi \in B} |\psi\rangle \langle\psi| = \mathbb{1}$ of the basis to obtain

$$H_{\text{MB}} = \sum_{\psi, \psi' \in B} |\psi\rangle \langle\psi| H |\psi'\rangle \langle\psi'| \,, \tag{C.1}$$

where the creation and annihilation operators in $H$ naturally act on the many-body states in Fock space. It is, however, important to take into account the anticommutation relations of the fermionic operators and the antisymmetry of the fermionic Fock states when evaluating the matrix elements $\langle\psi| H |\psi'\rangle$.

## D   Effect of different parameters on the topological phase diagram

Figure 6 shows the area of the topological region in $\varphi_3$-$r$ space for $\varepsilon_i = \varepsilon$ ans $\Gamma_j = \Gamma$ as a function of $|\varepsilon|$. The area increases with decreasing $|\varepsilon|$ and is the largest for vanishing dot energies $\varepsilon = 0$.

In Figure 7, the Chern number is shown as a function of $\varphi_3$ and $r$ for symmetric parameters $\varepsilon_i = \varepsilon = \Gamma = \Gamma_i$ in (a), and in (b)-(d) for asymmetric parameters. The asymmetry leads to a deformation and separation of the topological phase in two regions but does not increase the area of the topological phase in respect of $\varphi_3$ and the inter-dot coupling $r$.

## E   Weyl points in the doublet sector

To simplify the search for topology in the doublet sector, we look for Weyl points, which are points where the gap between the GS and first excited state (FES) vanishes. For simplicity, we assume $B = 0$, which reduces dimension of the doublet Hamiltonian to 4, as well as $\varepsilon_i = \varepsilon$ and $\Gamma_i = \Gamma$. Having found non-trivial topology for the SP ground state under these assumptions,

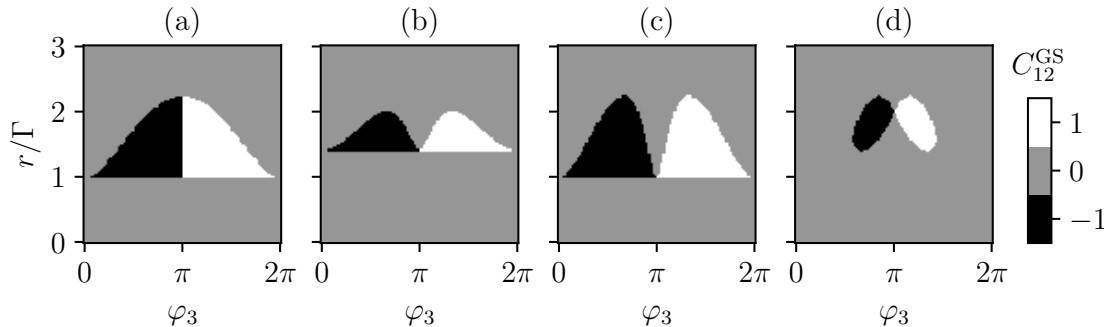

Figure 7: Topological phase diagrams for different sets of parameters. (a) for $\varepsilon_i = \varepsilon = \Gamma = \Gamma_i$. (b) for $\varepsilon_2 = 2\varepsilon_1 = \Gamma = \Gamma_i$. (c) for $\varepsilon_i = \varepsilon = \Gamma_0 = \Gamma_1$ and $\Gamma_2 = \Gamma_3 = 2\Gamma_0$. (d) for $\varepsilon_i = \varepsilon = \Gamma_0 = \Gamma_1 = \Gamma_2$ and $\Gamma_3 = 2\Gamma_0$.

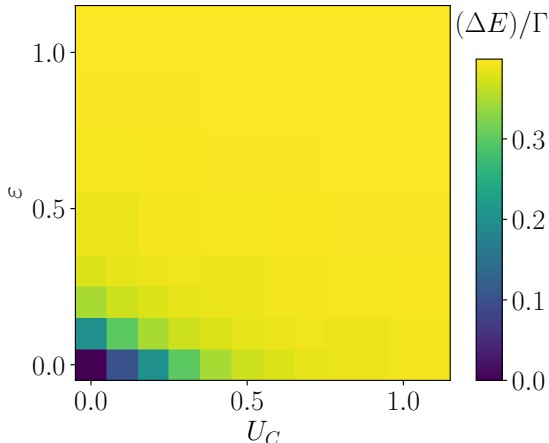

Figure 8: The energy gap $(\Delta E)/\Gamma$ between the GS and FES of the doublet sector minimized over all $\varphi_i \in [0, 2\pi]$ and $r \in [0.2\Gamma, 3\Gamma]$ for $B = 0$, $\Gamma = \Gamma_i = 1$ and different dot energies $\varepsilon = \varepsilon_i$ and Coulomb energies $U_C$.

we expect this to be no big limitation for topology of the doublet sector in the MB picture. Finally, we minimize the energy gap $(\Delta E)/\Gamma$ over all SC phase differences $\varphi_i$, $i = 1, 2, 3$ and $0.2\Gamma < r < 3\Gamma$. We exclude small inter-dot couplings $r < 0.2\Gamma$ here, since we do not expect any non-trivial topology is this case of quasi-decoupled subsystems. The resulting plot in Figure 8 shows a gap closing only at $\varepsilon = U_C = 0$ which is a potential candidate for a topological phase transition. However, the Chern number around that point remains zero, revealing no such phase transition.

# F  Effect of Coulomb interaction on the energy gap

Figure 9 clearly shows that the energy gap is larger in the presence of Coulomb repulsion on the quantum dots. In particular, the gap almost vanishes if $U_C = 0$ and $\varepsilon = 0$, but remains finite ($> 0.25\Gamma$) in a large parameter region if $U_C = 0.3\Gamma$ and $\varepsilon = 0$. Therefore, a finite Coulomb interaction may allow the measurement of a the quantized transconductance in the regime where the dot energies $\varepsilon$ approach the Fermi energies $E_F = 0$ of the SCs.



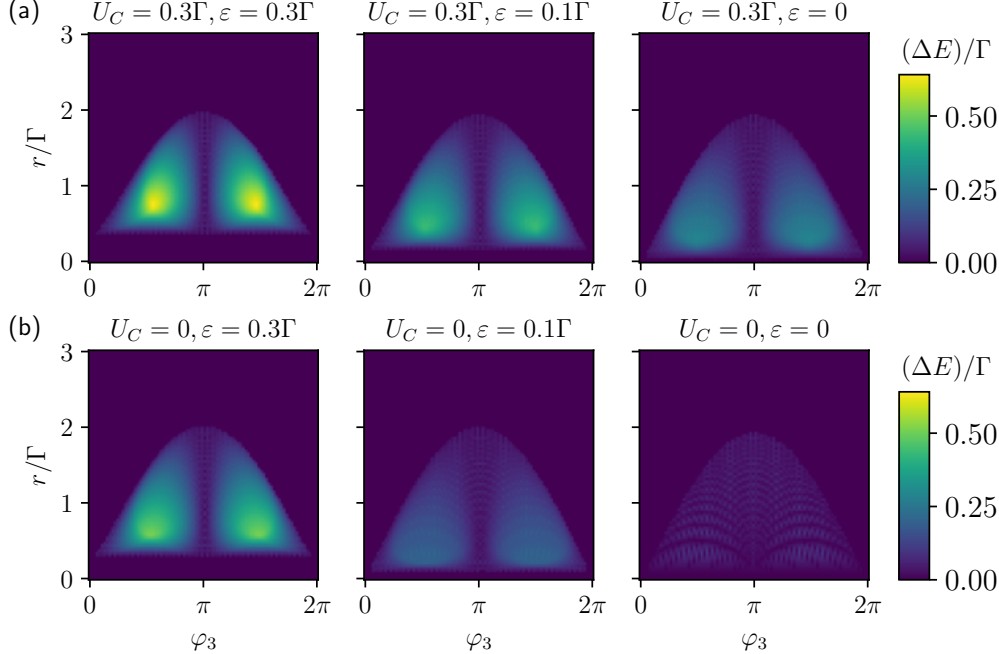

Figure 9: The energy gap $(\Delta E)/\Gamma$ between the GS and FES of the singlet sector (a) for finite Coulomb interaction $U_C = 0.2\Gamma$ and different $\varepsilon$, (b) without Coulomb interaction and the same $\varepsilon$ as above.

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
