# Peer review of "Ground state topology of a four-terminal superconducting double quantum dot"

_SciPost Physics, doi:SciPost Phys. 15, 214 (2023)_

## Round 1 · Referee Report · Anonymous (Referee 2) · 2023-5-30

Strengths

1 - A nice, relatively compact model with easily-interpretable parameters that exhibits parametric topology.
2 - Theoretical treatment of a configuration that is potentially accessible in experiments
3 - Starts to address the question of the influence of Coulomb interactions on Andreev parametric topology.

Weaknesses

1 - The size of the energy gap is hardly discussed, especially in regards to the interplay of charging energy and size of the topologically non-trivial region in parameter space.
2 - Missing explanation regarding the doublet states hosting no topologically nontrivial phase.
3 - References could be expanded or more carefully described.

Report

The authors describe a nice model for achieving non-trivial parameter-space topology of Andreev bound states in a four-terminal, double quantum dot structure. Overall, the results seem sensible to me. I have a few comments and concerns.

I was surprised that the doublet Hamiltonian block, a 4x4 matrix that depends on all phases, showed no nontrivial topology. It would be helpful if the authors could comment more on why this is. Was the full space of parameters explored? Do Weyl points appear, but only in pairs such that Chern number stays zero? Or, does the absence of topology in the doublet manifold imply a rotation into a further simplified form?

In addition to nonzero Chern number, any transconductance quantization will depend strongly on the size of the gap to the lowest excited state manifold. Thus, optimizing the area of the region in parameter space with nonzero Chern number is not by itself enough to claim an “enhancement”. If the added region has a tiny gap then it is not especially useful. It would be helpful if the authors could overlay the size of the topological gap in the nontrivial region and comment on if there is any tradeoff between the size of the region and the size of the gap in that region based on the way that they tune this.

Finally, one of the most relevant experimental contexts for dots coupled to superconductors involves Al-InAs superconductor-semiconductor heterostructures. With this materials platform, spin-orbit coupling is established as an important factor for mesoscopic devices. This work does not yet account for, nor does it even comment on, the potential impact of spin-orbit interactions. Since this would mix the singlet and triplet manifolds, it may be quite relevant for understanding the parametric topology of the device. I appreciate that extending the model to add spin-orbit may be beyond the desired scope of this manuscript, but this aspect at least deserves comment.

Requested changes

1 - A closer look at Equation 11. In particular, should the fourth diagonal entry be $2\epsilon_2 + U_C$?

2 - Discussion of the impact of the charging term on the topological gap energy.

3 - Deeper explanation for doublet states hosting no topologically nontrivial phase.

4 - References 27-29 do not involve Andreev levels, although the text implies such. Those references involve circuit plasma modes, which are qualitatively distinct from Andreev levels.

5 - This paper concerns itself with the effect of Coulomb interactions on Andreev levels but is rather sparing regarding citing the extensive publication history of this subject, including relatively recent works that address this subject.

6 - Remark on possible impact of spin-orbit interaction.

  • validity: high
  • significance: good
  • originality: good
  • clarity: high
  • formatting: excellent
  • grammar: good

Author:  Wolfgang Belzig  on 2023-09-04  [id 3950]

(in reply to Report 2 on 2023-05-30)

We thank the referee very much for the report. We addressed your requested changes in the following way:

1) This was a typo, thank you for pointing it out.

2) We now discuss the topological gap in section 2.5 and Appendix F. The effect of the dot energies $\varepsilon$ on the gap is shown in Figure 5 and the effect of the Coulomb interaction in Figure 9.

3) We did not findy any deeper explanation for the absence of doublet topology, but we solidified our claim by searching a large parameter region (last paragraph of section 2.3, Appendix E).

4) We corrected it.

5) We included additional citations [46-52] concerning Coulomb interaction in quantum dot junctions.

6) We commented on spin-orbit interaction in the last paragraph of section 3

---

## Round 1 · Referee Report · Anonymous (Referee 1) · 2023-5-30

Report

The occurence of robust Weyl crossings in the Andreev spectrum of 4-terminal Josephson junctions has attracted some interest since they were predicted in cited Ref. [13]. The main interest of the present work is to show that a double-dot structure, where each of the single-level dots interacts with two phase-biased superconducting terminals, can accommodate such Weyl crossings. The work first shows the occurence of Weyl points in the absence of Coulomb interaction, and in the limit of a large superconducting gap in the leads. Then it finds that the topological phases are robust in the presence of weak Coulomb interaction and/or a small Zeeman field.

These are encouraging results for the experimental realization of Weyl points, given that double-dot structures seem perhaps easier to realize than the specific setups addressed in earlier studies. On the other hand, I am a bit disappointed that the authors did not put more efforts in obtaining generic statements, while those presented in the manuscript may seem contingent to arbitrarily chosen sets of parameters. Before making a recommendation on the manuscript, I would thus like the authors to consider the following comments:

1) In the absence of interaction, the condition for Weyl points reduces to $c=0$ with $c$ of Eq. (7). One can notice that, at $\epsilon_1\epsilon_2>0$, $c$ is the sum of three squares: $c=(r^2-|\epsilon_1\epsilon_2|-|\gamma_1\gamma_2|)^2+(|\epsilon_1\gamma_2|-|\epsilon_2\gamma_1)^2+4r^2|\gamma_1\gamma_2|\cos^2(\psi/2)$, where $\gamma_1=\Gamma_0+\Gamma_1e^{i\phi_1}$, $\gamma_2=\Gamma_2e^{i\phi_2}+\Gamma_3e^{i\phi_3}$, and $\psi=\arg(\gamma_1\gamma_2^*)$. With this observation, it should becomes easy to make more general statements on the space of parameters that allows stabilizing the topological phase, as well as explain the location of Weyl points in phase space, and the shape of the phase diagram shown in Fig. 3.

2) The relation between the Chern number computed in the non-interacting case, using the BdG Hamiltonian (5), and the one computed in the presence of interactions, using the many-body Hamiltonian (11) in the singlet sector, should be clarified. Their eventual correspondance at vanishing interaction $U_c\to 0$, should be discussed.

3) Statements such as the absence of phase dependence of one of the 5 energy levels in the Hamiltonian (11) projected in the singlet sector, or the topologically trivial character of the doublet states (despite their phase dependence) should be clarified.

4) I would also be more cautious about the claimed optimal region of device parameters that allows enlarging the topological region in page 8, given the absence of a systematic procedure to address the large range space of parameters that characterize the double-dot structure (except if a systematic procedure following a similar path as the one hinted to in point 1) above can be found).

Finally, I would like to point out that the Introduction contains an inaccuracy about the citations on the theory of topological superconducting circuits, Refs. [13-29], given that only a subset of them concern Andreev bound state physics.

  • validity: -
  • significance: -
  • originality: -
  • clarity: -
  • formatting: -
  • grammar: -

Author:  Wolfgang Belzig  on 2023-09-04  [id 3949]

(in reply to Report 1 on 2023-05-30)
Category:
answer to question
correction

We thank the referee very much for the nice and constructive report. We addressed the comments in the following way:

1) Thank you for the suggestion. It was indeed possible to solve the Weyl point conditions in the symmetric case $\Gamma_j = \Gamma$ and $\varepsilon_i = \varepsilon$. This yields the exact conditions for the appearance of a non-trivial topological region an explains the shape of the phase diagram (which we inlcuded as a dashed line in Figure 3(a)).

2) For $U_C = 0$, the Chern number of the many-body ground state coincides with the one of the single-particle ground state, because then both Hamiltonians describe the same physical system. We now mention this explicitly in section 2.3 Many-body Hamiltonian.

3) We did not find deeper explanation (based on symmetry arguments) for the topological triviality of the doublet sector. We did, however, search a large parameter region to support our claim. Instead of looking directly at the Chern number, which is computationally expensive, we instead looked for parameters that allowed for a closing of the gap between ground state and first excited state of the doublet sector. We found this to be the case only for $\varepsilon = U_C = 0$ but did not find changes of the Chern number in the vicinity of this point (see Appendix E).

4) We agree that our statement was maybe a little bit bold. Now, we have have an exact solution for the symmetric case in the single-particle picture and showed at least for some asymmetric parameters (Figure 7) that the topological region is not enlarged. In the many-body picture, we argue that a Coulomb interaction in the allowed range ($U_C < 0.5\Gamma$) has only a small effect on the phase diagram and we can, therefore, resort to our findings in the single-particle picture.

5) We changed the formulation to avoid the misunderstanding with the references.

---

## Round 2 · Referee Report · Anonymous (Referee 2) · 2023-9-11

Report

After reviewing the requested changes by both of the previous reports and comparing with the updated version of the manuscript, I believe all the noted technical issues have been addressed.

As far as criteria/expectations for publishing, this work easily meets the criteria of SciPost Physics Core. The work provides a clear and simple model of some physics that is well-developed by their group and others, so I am less certain about publishing in SciPost Physics, where a high degree of innovation is typically required.

---

## Round 2 · Referee Report · Anonymous (Referee 1) · 2023-10-21

Report

In my opinion the authors answered satisfactorily to all remarks from the reviewers. Their results are original. They provide the possibility to discuss the effect of interactions on the topological properties of superconducting multiterminal systems made with quantum dots. I believe the article can be inspirational for future theoretical and experimental work in the field. Therefore, I recommend its publication in Sci Post Physics.

---

## Round 2 · Author Response

Dear Editor,

hereby we resubmit our article to scipost. We have addressed all comments by the reviewers. Thank you and the reviewer for the time to evaluate the article. Since the referees were overall positive and in favour of publication we hope the article can be accepted for SciPost Physics.

Kind regards,
Wolfgang Belzig (on behalf of all authors)

---

## Round 2 · List of Changes

The list of changes is given in the replies to the referees.

---

## Editorial Decision

published